# Phase Target-Based Calibration of Projector Radial Chromatic Aberration for Color Fringe 3D Measurement Systems

**DOI:** 10.3390/s22186845

**Published:** 2022-09-09

**Authors:** Yuzhuo Zhang, Yaqin Sun, Nan Gao, Zhaozong Meng, Zonghua Zhang

**Affiliations:** Intelligent Photoelectric Multi-Dimensional Sensing Laboratory, Hebei University of Technology, Tianjin 300400, China

**Keywords:** 3D measurement, chromatic aberration of projector, chromatic aberration correction

## Abstract

The camera and projector are indispensable hardware parts of a color fringe projection 3D measurement system. Chromatic aberration between different color channels of the projector and camera has an impact on the measurement accuracy of the color fringe projection 3D profile measurement. There are many studies on camera calibration, but the chromatic aberration of the projector remains a question deserving of further investigation. In view of the complex system architecture and theoretical derivation of the traditional projector radial chromatic aberration method, a phase target based on projector radial chromatic aberration measurement and the correction method are proposed in this paper. This method uses a liquid crystal display with a holographic projection film as the phase target. The liquid crystal display sequentially displays red, green, and blue horizontal and vertical sinusoidal fringe images. The projector projects red, green, and blue horizontal and vertical sinusoidal fringe images to the phase target in turn, and calculates the absolute phases of the display fringes and reflection fringes, respectively. Taking the green channel as the reference channel, a phase coordinate system is established based on the phases of the vertical and horizontal directions displayed on the display screen, using the phase of the reflection fringes on the display screen as the ideal phase value of the phase point. Then, the phase coordinate system of the red and blue channels is transferred to the green phase coordinate system to calculate the chromatic aberration of the red-green channels and the blue-green channels, and pre-compensation is conducted. Experimental results prove that this method can measure and calibrate the radial chromatic aberration of the projector without being affected by the image quality of the camera. The correction effect of this method is that the maximum chromatic aberration of the red-green channel decreases from 1.9591/pixel to 0.5759/pixel, and the average chromatic aberration decreases from 0.2555/pixel to 0.1865/pixel. In addition, blue-green channel maximum chromatic aberration decreased from 1.8906/pixel to 0.5938/pixel, and the average chromatic aberration decreased from 0.2347/pixel to 0.1907/pixel. This method can improve the projection quality for fringe projection 3D profile measurement technology.

## 1. Introduction

With the development of modern digitization and informatization technologies, the requirements for the measurement accuracy of optical systems increase accordingly. Optical three-dimensional shape measurement technology is used in many fields, such as machine vision, industrial inspection, reverse engineering, and medical imaging, which all have a wide range of applications [1,2,3,4]. The fringe projection measurement technology has the advantages of non-contact, low cost, fast speed, and simple operation [5,6]. As the projection display device in the optical system, the digital projector directly affects the quality of the image collected by the camera, and thus the accuracy of the measurement system. Among them, the digital projector has also attracted extensive research due to its flexible programming and low cost [7]. The object to be measured usually has extremely rich colors. For colored objects, the most important thing is to be able to restore the three-dimensional appearance of the object while also restoring its own color information. The phenomenon that the different refractive indexes of the projector lens for different colors of light cause different colors of light to be imaged at different positions is called chromatic aberration. The existence of chromatic aberration makes the edge of the color image projected by the projector produce different color stripes, which seriously affects the quality of the image and the accuracy of the three-dimensional profile measurement results. Therefore, in order to obtain high-precision three-dimensional measurement results, the radial chromatic aberration problem of the projector needs to be solved in advance.

The methods for measuring and correcting radial chromatic aberration are divided into hardware-based methods and software-based methods. The hardware-based method generally uses a hardware system to change the imaging position of the red, green, and blue wavelength light [8,9,10]. For example, using an apochromatic lens to focus three wavelengths on the same plane [9], although the apochromatic lens automatically minimizes the residual chromatic aberration in the wavelength range and reduces the radial chromatic aberration, the residual error of the apochromatic lens is relatively large; the active lens control system corrects the chromatic aberration of the system by adjusting the distance between the image plane and the lens [10]. The system is cumbersome, and the preparation process is complicated. In general, the hardware-based method is not used. This article also uses a software-based method to correct chromatic aberration. The software-based method generally establishes a chromatic aberration model by analyzing the radial chromatic aberration distribution of the system to compensate for the radial chromatic aberration. At present, there are some methods to correct chromatic aberration based on software, but there is still room for improvement. Huang et al. proposed a method for compensating the analysis and compensation of radial chromatic aberration in a color-coded structured light three-dimensional measurement system based on projection transformation [11]. By comparing the printed circle on the whiteboard with the red, green, and blue projections of the projector, the coordinate position of the color projection circle on the image coordinate system of the projector is used to analyze and compensate the radial chromatic aberration of the projector. This method introduces the problem of color crosstalk and the radial chromatic aberration of the camera in the analysis process. In the compensation process, because the points less than the pixel points are selected in advance for chromatic aberration measurement, the accuracy is not high enough. The method in this paper is to correct the chromatic aberration of all the pixels projected by the projector, meaning that the accuracy is higher. Jordi et al. calibrate the radial chromatic aberration of the system by placing a white calibration plate in front of the camera and projector [12]. However, the error caused by LCA changed with the position change in the captured image, but this was not considered in their research. Indeed, this method does not consider the influence of camera error on the result; however, the method in this paper uses an LCD screen to eliminate the result of chromatic aberration correction of the projector by the camera. Zhang [13] and others proposed a linear compensation method to compensate for the radial chromatic aberration caused by the projector and the color charge-coupled device (CCD) camera lens. This method involves projecting different single-color fringe patterns with the same number of fringes. The difference between the calculated absolute phase images can be used to measure the radial chromatic aberration and determine the fringe aberration. It is suitable for measuring objects with discontinuities and large slopes. It is only suitable for the best multi-fringe selection method, and not suitable for the general stripe projection system. The method in this paper involves first calculating the chromatic aberration of each phase point projected by the projector, and then re-generating a new fringe pattern, so no matter which fringe projection system, a new fringe pattern can be generated according to the pixel difference of each point for the next experiment. Generally speaking, the cost of hardware-based methods is much higher than that of software-based methods, and the existing radial chromatic aberration methods basically calibrate the radial chromatic aberration of the system or camera, and not all of them are suitable for the radial chromatic aberration of the projector calibration. In this paper, a phase target-based calibration of projector radial chromatic aberration for color fringe 3D measurement systems is proposed. According to the problems mentioned above, the method provided in this paper can correct the global chromatic aberration of projector, with higher accuracy. Moreover, it can avoid the interference caused by camera error and have wider applicability. At the same time, this method is simple and feasible, without a cumbersome hardware system, and the experimental method and calculation process are not complicated. In the next section, the basic principle of phase target-based calibration of projector radial chromatic aberration for color fringe 3D measurement systems, the specific program flow, and the experimental process are introduced in detail.

## 2. Basic Principles

### 2.1. Color Fringe Projection 3D Measurement System

The workflow of the whole color fringe 3D measurement system is shown in Figure 1. During the measurement, firstly, the standard sinusoidal fringes generated by the computer are projected on the measured object through the digital light programming (DLP) projector. When a standard sinusoidal fringe grating is projected onto an object surface, the standard fringe will deform due to the uneven surface of the object. After deformation, the fringe grating is captured by the CCD camera, and then transmitted back to the computer. After a series of phase calculations, the wrapped phase diagram is obtained, and then the unwrapped phase diagram is obtained by some phase unwrapping algorithms. Finally, the three-dimensional coordinates of each point on the surface of the object are obtained using the system parameters calibrated in advance. To measure every direction of the whole object, it is necessary to convert different angles of the object. Then, the above steps are repeated to combine all directions together, and finally the overall three-dimensional profile of the object is obtained.

### 2.2. Phase Calculation Method

There are two steps to obtain the absolute phase of the measured object: Firstly, the wrapped phase information of the sinusoidal fringe image is extracted. Secondly, the wrapped phase is unwrapped to acquire phase information of the images. Two common methods of calculating the wrapped phase data are multi-step phase-shifting [14] and transform-based algorithms [15]. Although a transform-based algorithm can extract the wrapped phase from a single fringe pattern, it is time consuming and acquires less accurate phase data. Therefore, the four-step phase-shifting algorithm is used to accurately calculate the wrapped phase data [16]. To obtain the absolute phase map, many spatial and temporal phase unwrapping algorithms have been developed [17,18,19,20,21,22,23,24]. In this paper, the wrapping phase is calculated by a four-step phase shift algorithm, and the unwrapping phase is calculated by optimum 3-frequency selection.

The basic principle of the four-step phase shift algorithm is: firstly, four sinusoidal fringe patterns with a phase shift of 90 degrees are generated by a computer, such as 0, π/2, π, and 3π/2. Secondly, the projector projects these fringe patterns onto the surface of the measured object. Finally, the camera collects the deformed fringe patterns reflected by the measured object.

The light intensity of the sinusoidal fringe images collected by the camera at the t-th image point (*x,y*) is:(1)$$I\left(x,y\right)=I_{0}\left(x,y\right)+I_{00}\left(x,y\right)\cos\left[\varphi \left(x,y\right)+\frac{\pi }{2}(t-1)\right]+I_{n}(x,y)$$

In the formula, *I*_0_(*x,y*) is the value of background light intensity at point (*x,y*), which is related to ambient light and object surface; *I*_00_(*x,y*) is the amplitude of the sinusoidal fringe, which represents the fringe modulation degree; *φ*(*x,y*) is the measured phase value to be obtained; and *I_n_*(*x,y*) represents the influence of random noise on light intensity, which is generally ignored.

The wrapped phase at point (*x,y*) can be obtained:(2)$$\varphi \left(x,y\right)=\arctan\frac{I_{4}\left(x,y\right)-I_{2}\left(x,y\right)}{I_{1}\left(x,y\right)-I_{3}\left(x,y\right)}$$

When extracting the phase information from the image, the phase value of the wrapped phase is between −π/2 and π/2. For the whole pattern, the phase should be monotonically continuous, that is, each point corresponds to a unique phase value, so it is necessary to unwrap the wrapped phase.

Zhang and others put forward the method of optimum 3-frequency selection [18]. This method adopts three groups of phase shift fringe patterns of different frequencies, and the relationship between the three frequencies is as follows: N, N-1, N-√N. Taking the phase with fringe number N as the reference phase, the phases with stripe number N-1 and N-√N are subjected to difference frequency operation. Then, the fringe level of each pixel is calculated, and the phase unwrapping of the reference phase is carried out using the fringe order.

### 2.3. Projector Chromatic Aberration Analysis

Chromatic aberration refers to the phenomenon that light with different wavelengths cannot be focused on the same point in an optical lens. The reason for this is that the same lens has different refractive indexes for lights of different wavelength [8,25]. For light with a longer wavelength, the refractive index of the lens is lower. For light with a shorter wavelength, the refractive index of lens is higher. Therefore, red, green, and blue light will be imaged at different positions in the imaging system [26]. In the projector, because the projector projects divergent beams, the chromatic aberration at the imaging center is the smallest, and the farther away from the imaging center, the more obvious the chromatic aberration is.

Chromatic aberration is generally divided into lateral chromatic aberration and radial chromatic aberration. Chromatic aberration is caused by the different magnifications with different wavelengths of light in the lens, which leads to different imaging positions on the same imaging plane, that is, red, green, and blue colors are imaged at different positions on the same imaging plane, and some abnormal color stripes appear in the image. In general, the measurement error caused by lateral chromatic aberration can be reduced to a certain extent by using a lens with higher quality. In this paper, the investigation mainly focuses on the radial chromatic aberration.

## 3. Measurement Method of Radial Chromatic Aberration

In this paper, a method of phase target-based calibration of projector radial chromatic aberration for color fringe 3D measurement systems is proposed. The radial chromatic aberration of a projector is that the same projection point is imaged in different positions in the red, green, and blue channels. The final ideal effect of correction is that the three-color channels can be imaged at the same position by the correction method. The schematic diagram of radial chromatic aberration measurement is shown in Figure 2. In this paper, the green channel is regarded as an ideal channel, and the red channel and the blue channel are calibrated to the green channel.

The displayed fringe phase of liquid crystal display (LCD) includes the radial chromatic aberration of the camera. The reflected fringe phase of LCD includes the radial chromatic aberration of the camera and the projector. The flow chart of projector radial chromatic aberration calibration is shown in Figure 3.

Next, take the red and green chromatic aberration as an example to illustrate the specific method of calculating the radial chromatic aberration of projector, and the calibration method of blue and green chromatic aberration is the same. Because the green channel is regarded as an ideal channel, φ_Lcd_g_h_ and φ_Lcd_g_v_ should be taken as ideal phases to calibrate the chromatic aberration between the red channel and green channel. The final ideal effect of correction is that the red channel phase φ_Pro_r_h_ and φ_Pro_r_v_ projected by the projector is the same as the ideal green channel phase φ_Pro_g_h_ and φ_Pro_g_v_ projected by the projector. The specific process is as follows:The camera captures the horizontal and vertical sinusoidal fringe images of green and red displayed by LCD and calculates their absolute phases φ_Lcd_g_h_, φ_Lcd_g_v_ and φ_Lcd_r_h_, φ_Lcd_r_v_.The camera captures the horizontal and vertical sinusoidal fringe images of red and green projected by the projector, and calculates their absolute phases φ_Pro_r_h_, φ_Pro_r_v_ and φ_Pro_g_h_, φ_Pro_g_v_. All phases are obtained by the four-step phase shift algorithm and the optimum 3-frequency selection method.According to the above four absolute phase diagrams of red and green, the phase coordinate system is established with φ_Lcd_g_v_ as the abscissa, φ_Lcd_g_h_ as the ordinate, and φ_Pro_g_h_ and φ_Pro_g_v_ as the amplitude, and the absolute phases of the red channel and green channel are unified into this phase coordinate system.Because of the camera error, φ_Lcd_g_h_, φ_Lcd_g_v_ and φ_Lcd_r_h_, φ_Lcd_r_v_ in the same position are different. In the green phase coordinate system, take the phase value of the red phase point projected by the projector when φ_Lcd_r_h_ and φ_Lcd_r_v_ is the same as that of φ_Lcd_g_h_ and φ_Lcd_g_v_. At this time, the ideal phase value of the red channel is φ_Pro_r_h_ideal_ and φ_Pro_r_v_ideal_. As shown in Figure 4, there are two phase points A_G_ and A_R_ in the green fringe phase displayed by LCD at point A, due to the camera chromatic aberration. After fitting through the above process, the positions of the red and green phase points coincide, that is, B_G_ = A_G_. In the fringe phase reflected by LCD, the corresponding ideal projection phase points of red and green are B_G_ and 0.

When the red channel phase point and the green channel phase point projected by the projector are located at the same position, the phase difference is called chromatic aberration. The calculation methods (formulas) of horizontal and vertical radial chromatic aberration are as follows:


(3)
$$\Delta \varphi_{RG\_h}=\varphi_{\mathrm{Pr}o\_r\_h\_ideal}-\varphi_{\mathrm{Pr}o\_g\_h}$$



(4)
$$\Delta \varphi_{RG\_v}=\varphi_{\mathrm{Pr}o\_r\_v\_ideal}-\varphi_{\mathrm{Pr}o\_g\_v}$$



(5)
$$\vec{\Delta \varphi_{RG}}=\left(\Delta \varphi_{RG\_v},\Delta \varphi_{RG\_h}\right)$$


Similarly, the radial chromatic aberration between the blue channel and green channel in horizontal and vertical directions is calculated as follows:(6)$$\Delta \varphi_{BG\_h}=\varphi_{\mathrm{Pr}o\_b\_h\_ideal}-\varphi_{\mathrm{Pr}o\_g\_h}$$
(7)$$\Delta \varphi_{BG\_v}=\varphi_{\mathrm{Pr}o\_b\_v\_ideal}-\varphi_{\mathrm{Pr}o\_g\_v}$$
(8)$$\vec{\Delta \varphi_{BG}}=\left(\Delta \varphi_{BG\_v},\Delta \varphi_{BG\_h}\right)$$

## 4. Correction of Radial Chromatic Aberration

The ideal phases φ_Pro_r_h_ideal_, φ_Pro_r_v_ideal_ and φ_Pro_g_h_ideal_, φ_Pro_g_v_ideal_ of the red channel and the blue channel are obtained from the calibration process of projector radial chromatic aberration in Section 3. The correction process involves getting the pre-compensated phase of the red channel and the blue channel. According to the calculated pre-compensation phase, a new fringe is regenerated, and then the new fringe is re-projected. The correction process is shown in Figure 5.

The specific process is as follows:1.Calculate the compensation phase of the projector

The ideal vertical and horizontal phases of the fringes reflected by LCD at BG and BR in Figure 6 are ($\varphi_{Pro\_v\_R}^{B}$, $\varphi_{Pro\_h\_R}^{B}$), and ($\varphi_{Pro\_v\_R}^{B}$, $\varphi_{Pro\_h\_R}^{B}$)= ($\varphi_{Lcd\_v\_G}^{B}$, $\varphi_{Lcd\_h\_G}^{B}$), in which C_R_’s coordinates ($\varphi_{Pro\_v\_R}^{C}$, $\varphi_{Pro\_h\_R}^{C}$) are the same phase point of BR in the red fringe phase reflected by LCD, and its corresponding coordinates of the red LCD displayed fringe phase are ($\varphi_{Lcd\_v\_R}^{C}$, $\varphi_{Lcd\_h\_R}^{C}$). In this process, the red channel is offset in advance, so that the image projected by the red channel coincides with that projected by the green channel; thus, achieving the purpose of correction.

2.Matching the pixel value of the projector resolution

The pre-compensated phase of the projector obtained in the first step is compensated under the resolution of the CCD camera. At present, the calculated chromatic aberration is obtained from the image taken by the camera, and the new fringe image needs to be re-projected by the projector. Since the resolution of the projector is different from the resolution of the CCD camera, the compensated phase should be converted to the pixel under the resolution of the projector.

The mapping relationship between the projector pixel coordinate system and the projector phase coordinate system is established by using the ideal horizontal and vertical phase shift fringe sequence generated by computer. Assuming that the absolute phase of a point M in the pixel coordinate system of the CCD camera is $\varphi_{h}^{C}$(u^c^,v^c^) in the horizontal direction and $\varphi_{v}^{C}$(u^c^,v^c^) in the vertical direction, and its corresponding pixel coordinate in the resolution of the projector is M(u^p^,v^p^), the relationship between the pixel in the resolution of the projector and the phase of the pixel coordinate system of the CCD camera is:(9)$$\{\begin{array}{c} v^{p}=\frac{H\varphi_{h}^{C}\left(u^{c},v^{c}\right)}{2\pi T}+\frac{H}{2}\\ u^{p}=\frac{V\varphi_{v}^{C}\left(u^{c},v^{c}\right)}{2\pi T}+\frac{V}{2} \end{array}$$
in which *H* is the width of projected fringe image, *V* is the height of projected fringe image, and *T* is the maximum number of projected sinusoidal fringes.

Because the resolution of the CCD camera is inconsistent with that of the projector, and there is a certain included angle between the optical axes, the pixels of the CCD camera and the projector do not correspond to each other one by one, so the integer value should be obtained by two-dimensional interpolation.

3.Matching fringe brightness

The horizontal and vertical pixel coordinates of the reprojected red channel and blue channel can be obtained from step 2 above. Taking the red channel and the green channel as an example, the initial projection of the green channel pixel matrix and its brightness matrix are mapped one by one in the same pixel position. The re-projected green channel pixel matrix and its brightness matrix are mapped one by one at the same pixel position. The initial projection pixel and its brightness matrix are generated by computer software, and the reprojection pixel matrix can be obtained by the above step 2. As shown in Figure 7, point P in the reprojected pixel matrix is taken. The phase point $A_{R}^{P}$ in the red channel is different from the phase point $A_{G}^{P}$ in the green channel. $A_{R}^{P}$(u,v) is the offset pixel value that needs to be pre-compensated in the green channel for the calculated red channel. Firstly, the initial projection pixel matrix and the initial projection brightness matrix are interpolated synchronously. Then, the in-phase point $B_{R}^{P}$ of the point $A_{R}^{P}$ in the interpolated pixel matrix is obtained. The interpolation method used in this process is the grid data interpolation function. Secondly, the mapping relationship of the pixel matrix and the brightness matrix with the same pixel position is used to obtain the brightness $I_{B}^{R}$ of point $B_{R}^{P}$ in the brightness interpolation matrix. Finally, the brightness at point $A_{R}^{P}$ is $I_{A}^{R}$(u,v) = $I_{B}^{R}$(u,v). By using this method, the brightness of the pre-compensated images (2 × 4 groups × 3 pictures) of the four-step phase shift sequence in accordance with the frequency in the horizontal and vertical directions of the red channel in the green channel are calculated. In the same way, the offset brightness value of the blue channel is $I_{A}^{B}$.

4.The process of re-projection

Using the pre-compensation phase of red and blue, through the process of pixel and brightness matching, red and blue pre-compensated fringe images can be generated. Moreover, the new fringe images are in accordance with the requirements of the four-step phase shift algorithm and the optimum 3-frequency selection method. Then, re-projecting the new horizontal and vertical fringe images, the CCD camera can capture the new fringe images displayed by LCD and reflected by LCD, and the computer can calculate their unwrapped phases again. The fringe image of the green channel does not change, and the green color is still used as the reference color. Then, when the red channel and the blue channel are in the same phase as the green fringes displayed by LCD, the difference between the fringe phase reflected by LCD of the red and blue channels and the fringe phase reflected by LCD of the green channel can be compared.

## 5. Experiment Results and Analysis

The experimental system is shown in Figure 8, which consists of a CCD (charge coupled device) color camera, LCD (liquid crystal display), DLP (digital light processing) projector and a computer.

The LCD with holographic projection film is used as the phase target, and the Moire fringe can be eliminated in the process of LCD fringe image collection by the camera. The target plane of the projector is parallel to the phase target, and all the pixel points of the projector can be imaged on the LCD plane. The phase target is within the depth of field of the CCD camera and the DLP projector. The CCD camera and LCD are connected to the computer through the Ethernet port and DP port, respectively, to realize the display, collection, and preservation of images. The camera model is a ECO424CVGE color camera with a resolution of 492 × 656. The projector model is CP270(BenQ) with a resolution of 1024 × 768.

Taking the green channel as the reference color and the horizontal direction as an example, as shown in Figure 9, “o” indicates the phase of the green channel, “*” indicates the red channel phase, and “×” indicates the blue channel phase. The fringe phase displayed by LCD and the fringe phase reflected by LCD of the RGB three-channel are unified into their respective fringe phase coordinate system displayed by LCD. Additionally, the distribution of some RGB channel phase points in the fringe phase coordinate system displayed by LCD are shown in Figure 9a. Figure 9b is the main view of Figure 9a. The positions of the red, green, and blue channels do not coincide, indicating that the projector has chromatic aberration. Figure 9c is the top view of Figure 9a. The positions of the red, green, and blue channels do not coincide, indicating that the CCD camera has chromatic aberration.

Taking the horizontal direction as an example, Figure 10 is a schematic diagram showing the distribution of 10 red and blue phase points. The distribution of phase points is obtained according to the method of calculating the ideal phase of red and blue color channels and the reference color channels in Section 3. Among them, the phase coordinates of the red, green, and blue channels are the same, but the amplitudes are different. Additionally, the amplitudes are the chromatic aberration between the red channel and green channel and the chromatic aberration between the blue channel and green channel.

The radial chromatic aberration of the projector is calculated by using the method of calculating horizontal and vertical radial chromatic aberration in Section 3. In order to show the distribution rule of the chromatic aberration between the red and green channels and between the blue and green channels in the green channels, the vector variation trend of the red−green chromatic aberration and the blue-green chromatic aberration were shown by taking finite points at intervals in the whole field of view. The chromatic aberration diagram of the red−green channel is shown in Figure 11. Additionally, the chromatic aberration diagram of the blue-green channel is shown in Figure 12. The length and direction of the arrow represent the value and direction at the phase point.

According to step 4, a pre−compensated fringe image is generated by using the matched luminance matrix, as shown in Figure 13.

After the reprojection, the green channel is still used as the reference, and the steps in Section 3 are repeated. Then, the phase point distribution between the red and green channels and between the blue and green channels after reprojection are compared, as shown in Figure 14. Figure 14a shows the calibration results of chromatic aberration of the red-green channel after reprojection. Figure 14b shows the calibration results of chromatic aberration of the blue-green channel after reprojection. Comparing the chromatic aberration before and after calibration, at the time of initial projection, the maximum chromatic aberration of the red-green channel is 1.9591/pixel, and the average value is 0.2555/pixel; the maximum chromatic aberration of the blue-green channel is 1.8906/pixel, and the average value is 0.2347/pixel. After re-projection, the maximum chromatic aberration of the red and green channels is 0.5759/pixel and the average value is 0.1865/pixel, while the maximum chromatic aberration of the blue and green channels is 0.5938/pixel and the average value is 0.1907/pixel. As shown in Figure 15, the black arrow represents the radial chromatic aberration calibration result after re-projection. Figure 15a shows the chromatic aberration vector variation between the red and green channels before and after calibration, and Figure 15b shows the chromatic aberration vector variation between the blue and green channels before and after calibration.

The traditional method and the proposed method are used to restore the three-dimensional shape of the steps and verify the measurement accuracy of the optical system before and after pre−compensation. Figure 16a shows the three-dimensional shape of the steps restored by the traditional method; Figure 16b shows the three-dimensional shape of the steps restored by the proposed calibration method.

Processing the 3D depth data before and after calibration, and calculating the distance between two planes of the step, Table 1 shows the measurement accuracy of the three-dimensional measurement system before and after calibration. The measurement error of the steps distance before and after calibration is reduced from 0.724 mm to 0.082 mm. The measurement accuracy of the optical measurement system is improved significantly, which verifies the feasibility and effectiveness of the calibration method.

In this paper, a machine part with large gradient steps is used to verify the effectiveness of the presented method. The accuracy of this method compared with the related work can be quantitatively evaluated by measuring the distance between steps. This method can effectively reduce the error of the measured distance between step 1 and step 2 from 0.727mm to 0.069mm after correcting the chromatic aberration of projector. The distance error between step 2 and step 3 is reduced from −0.721mm to −0.095mm. The error can be reduced by nearly 90%. The measurement accuracy is obviously improved. The verification results of this method are compared with the measurement results of other studies. Huang et al. proposed a method [11] for compensating the analysis and compensation of radial chromatic aberration in a color-coded structured light three-dimensional measurement system based on projection transformation. By comparing the printed circle on the whiteboard with the red, green, and blue projections of the projector, the coordinate position of the color projection circle on the image coordinate system of the projector is used to analyze and compensate the radial chromatic aberration of the projector. The method uses a high precision flat board to verify the effectiveness of the method. This method reduces the maximum error of the measuring plate from 0.258 mm to 0.110 mm, and the standard deviation from 0.092 mm to 0.028 mm; the error is reduced by more than half. Obviously, the method in reference [11] is not as accurate as the method mentioned in this research. Pagès et al. [12] employed a white flat panel, which was set in front of the camera and the projector to calibrate the LCA of the system. Three patterns, which include red, green, and blue fringes, respectively, were projected onto the panel and captured by the camera. The median value of relative positions of different color fringes was calculated for every scan line. The median values were used to compensate the difference of the positions between the green channel and the other two channels. The method also uses a high precision flat board to verify the effectiveness. It reduces the maximum error of the measuring plate from 0.258 mm to 0.142 mm, and the standard deviation from 0.092 mm to 0.036 mm. The error can be reduced by nearly half. By comparison, it can be concluded that the method of correcting the chromatic aberration of the projector proposed in this paper is more competitive in improving the measurement accuracy.

## 6. Conclusions

In this paper, a method for measuring and pre-compensating the radial chromatic aberration of a projector is proposed, which uses LCD to display the absolute phase of horizontal and vertical fringes of red, green, and blue, and uses green as the reference color to establish a phase coordinate system. In the process of radial chromatic aberration measurement, the fringe images displayed by LCD and the fringe images projected by the projector are captured by camera in the same pose state. The difference between the two sets of phases is whether there is radial chromatic aberration of the projector. The measurement and correction method of radial chromatic aberration proposed in this paper avoids the coupling error between camera distortion and projector distortion. Moreover, the projection quality of the projector and the measurement accuracy in the optical measurement system are improved. Moreover, this method does not need the process of system calibration, which makes the experimental process more flexible and easier to operate.

## Figures and Tables

**Figure 1 sensors-22-06845-f001:**
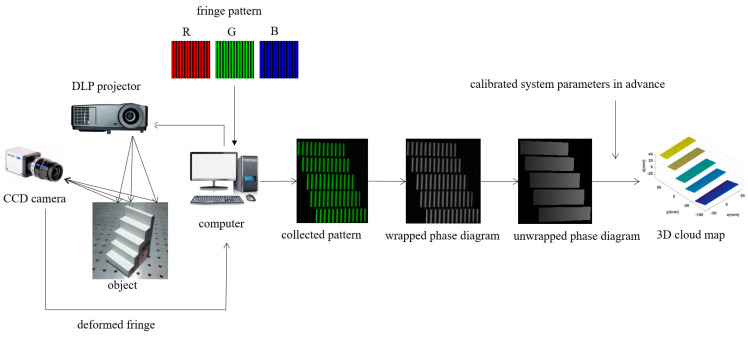
Workflow of the color fringe three−dimensional measurement system.(DLP: digital light programming; CCD: charge coupled device; LCD: liquid crystal display).

**Figure 2 sensors-22-06845-f002:**
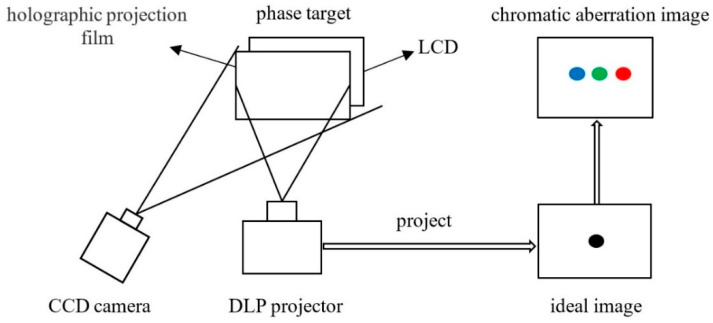
Schematic diagram of radial chromatic aberration measurement. (DLP: digital light programming; CCD: charge coupled device; LCD: liquid crystal display).

**Figure 3 sensors-22-06845-f003:**
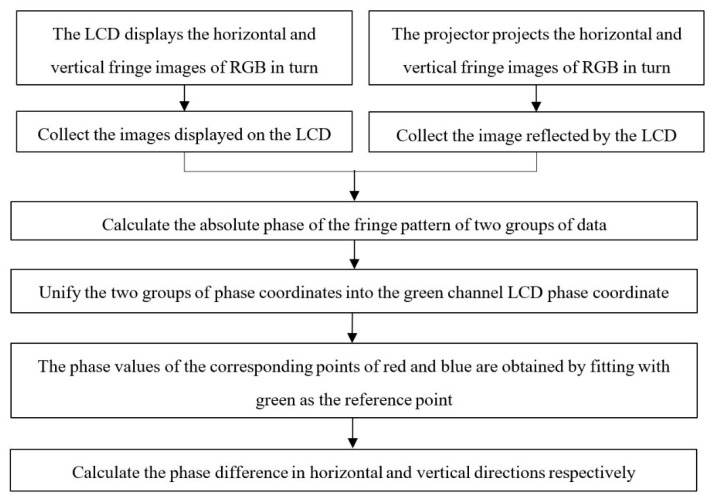
Flow chart of radial chromatic aberration calibration.

**Figure 4 sensors-22-06845-f004:**
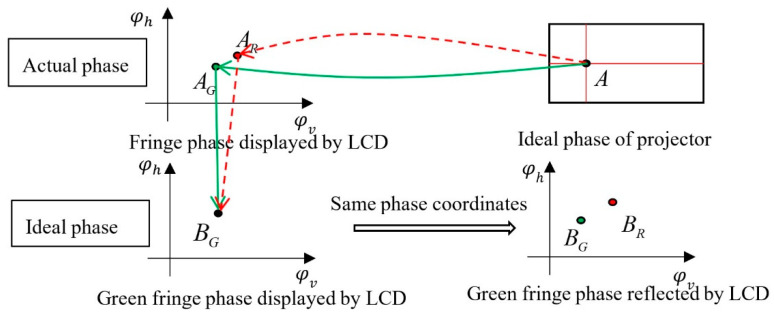
Schematic diagram of ideal phase of the red channel.

**Figure 5 sensors-22-06845-f005:**
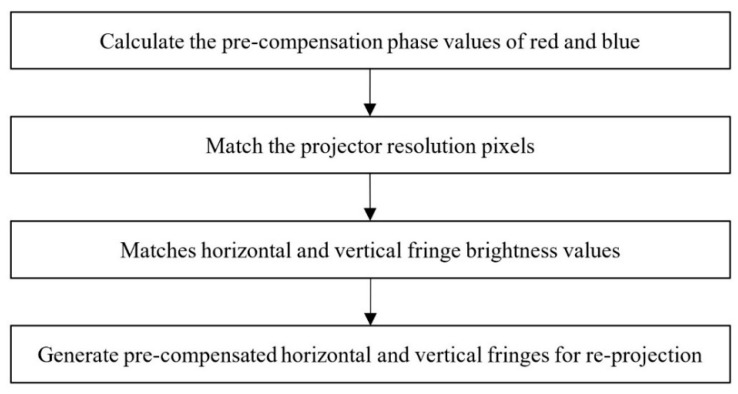
Flow chart of radial chromatic aberration correction.

**Figure 6 sensors-22-06845-f006:**
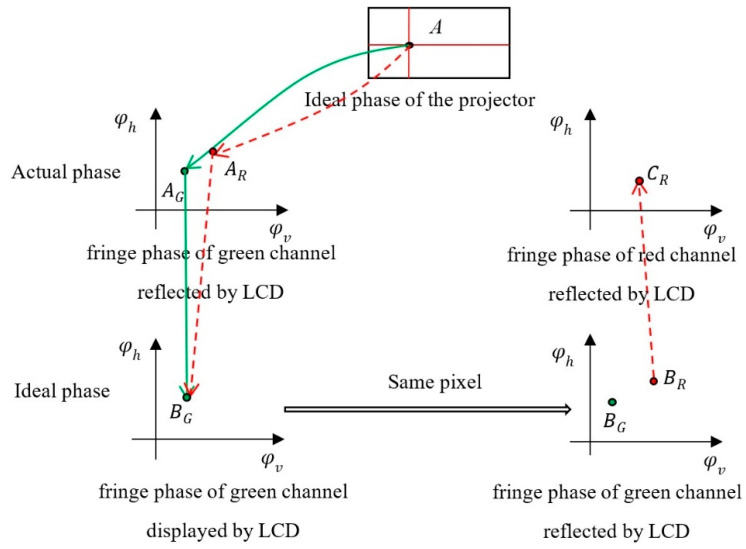
Phase mapping relationship of chromatic aberration compensation for the red channel.

**Figure 7 sensors-22-06845-f007:**
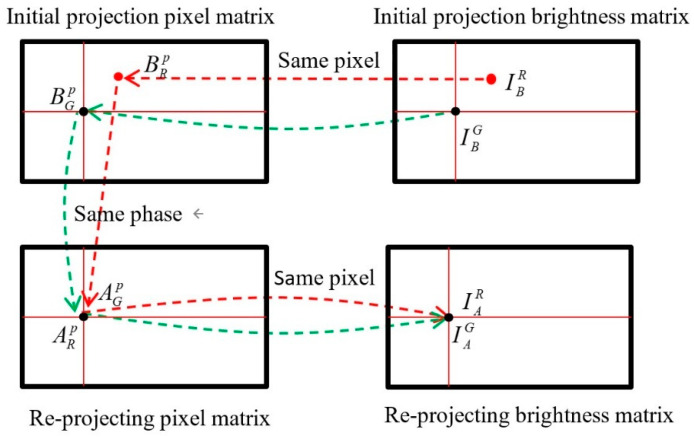
Schematic diagram of brightness matching.

**Figure 8 sensors-22-06845-f008:**
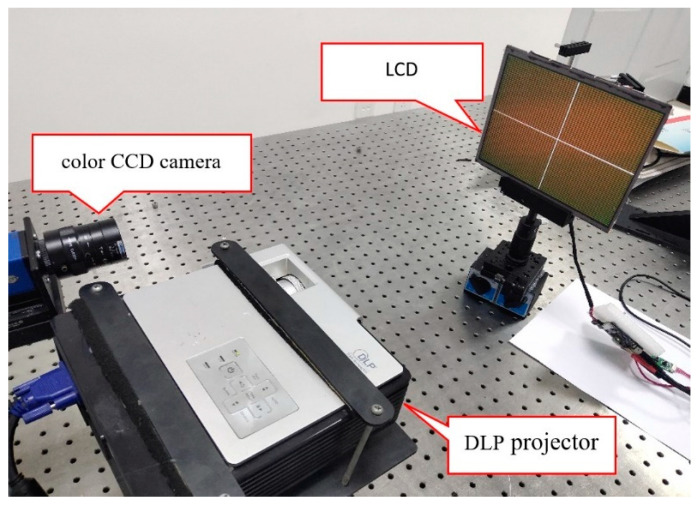
Hardware system diagram of the radial chromatic aberration experiment.

**Figure 9 sensors-22-06845-f009:**
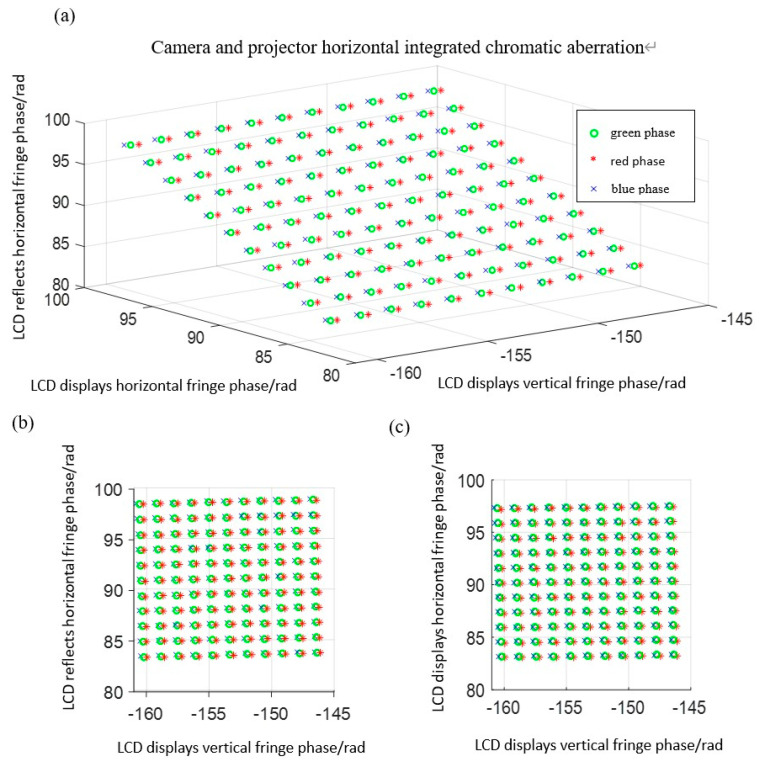
Comprehensive chromatic aberration of the camera and projector in the horizontal direction: (**a**) three−dimensional coordinate system diagram; (**b**) front view; (**c**) top view.

**Figure 10 sensors-22-06845-f010:**
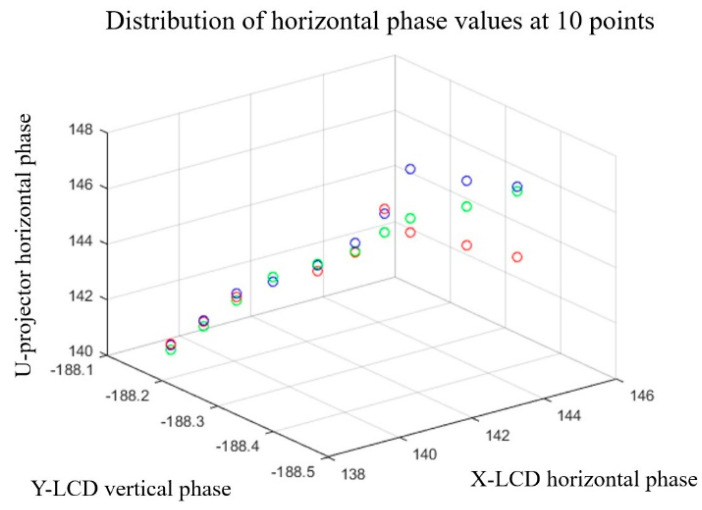
Schematic diagram of horizontal chromatic aberration of three−color channels.

**Figure 11 sensors-22-06845-f011:**
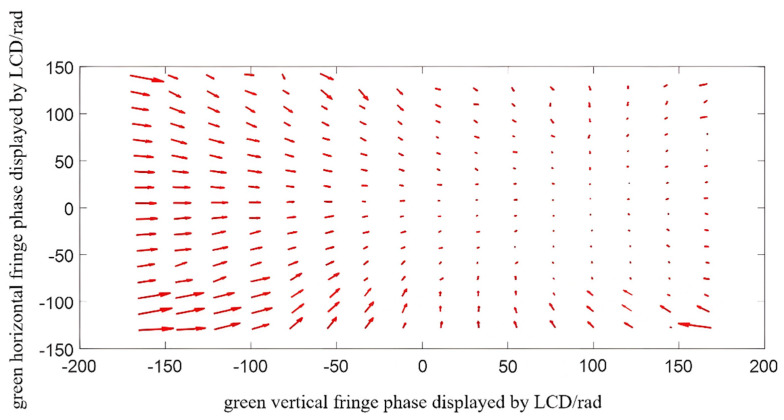
Schematic diagram of the chromatic aberration vector of the red and green channels.

**Figure 12 sensors-22-06845-f012:**
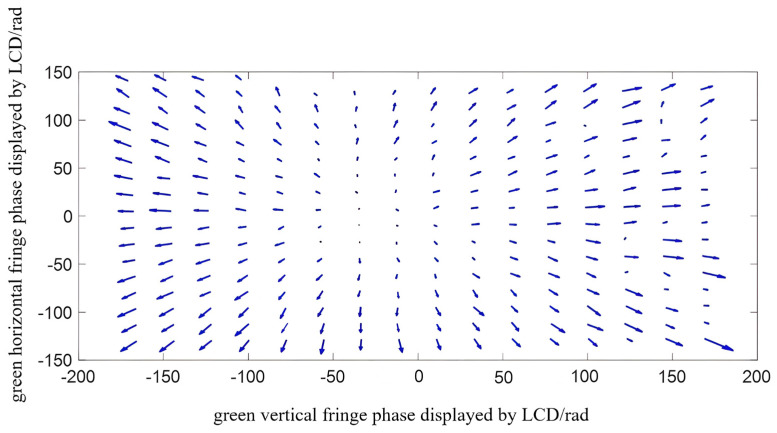
Schematic diagram of the chromatic aberration vector of the blue and green channels.

**Figure 13 sensors-22-06845-f013:**
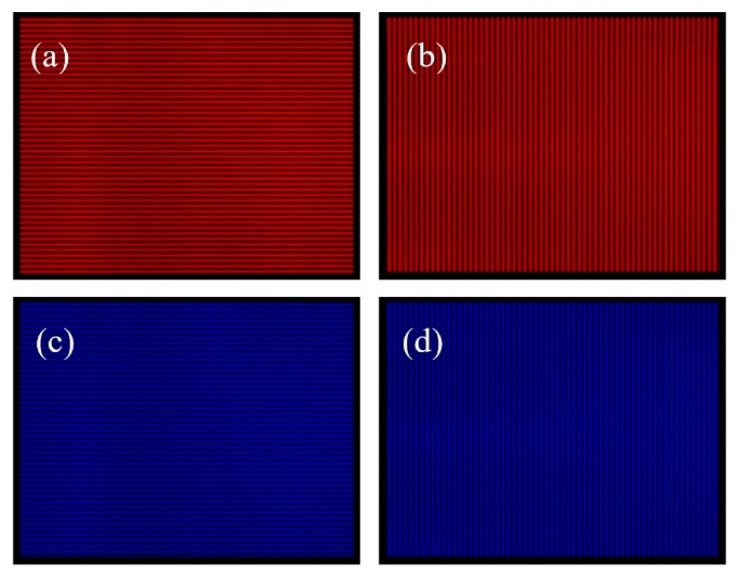
Pre-compensated fringe image: (**a**) red channel pre-compensated horizontal fringe image; (**b**) red channel pre-compensation vertical stripe image; (**c**) blue channel pre-compensation horizontal stripe image; (**d**) blue channel pre-compensation vertical stripe image.

**Figure 14 sensors-22-06845-f014:**
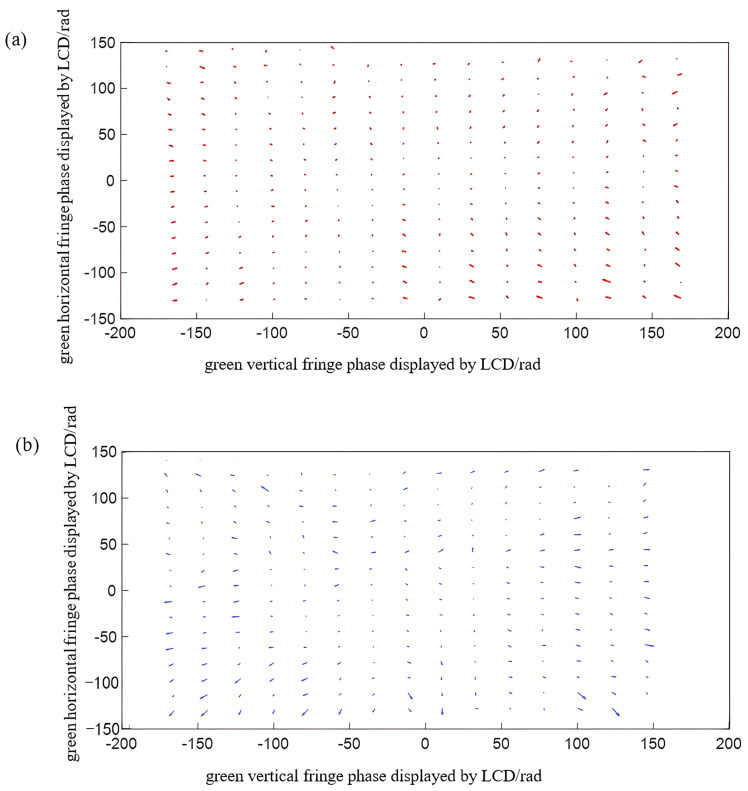
Radial chromatic aberration vector diagram of projector after re−projection: (**a**) vector change diagram of the red and green channels; (**b**) vector change diagram of the blue and green channels.

**Figure 15 sensors-22-06845-f015:**
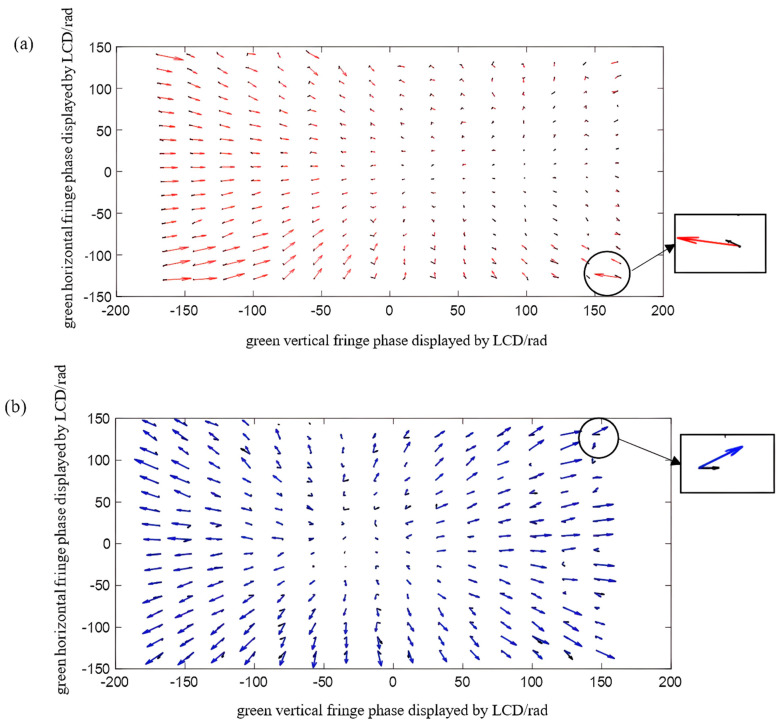
Projection radial chromatic aberration vector change diagram before and after pre-compensation: (**a**) vector change diagram of the red and green channels; (**b**) vector change diagram of the blue and green channels.

**Figure 16 sensors-22-06845-f016:**
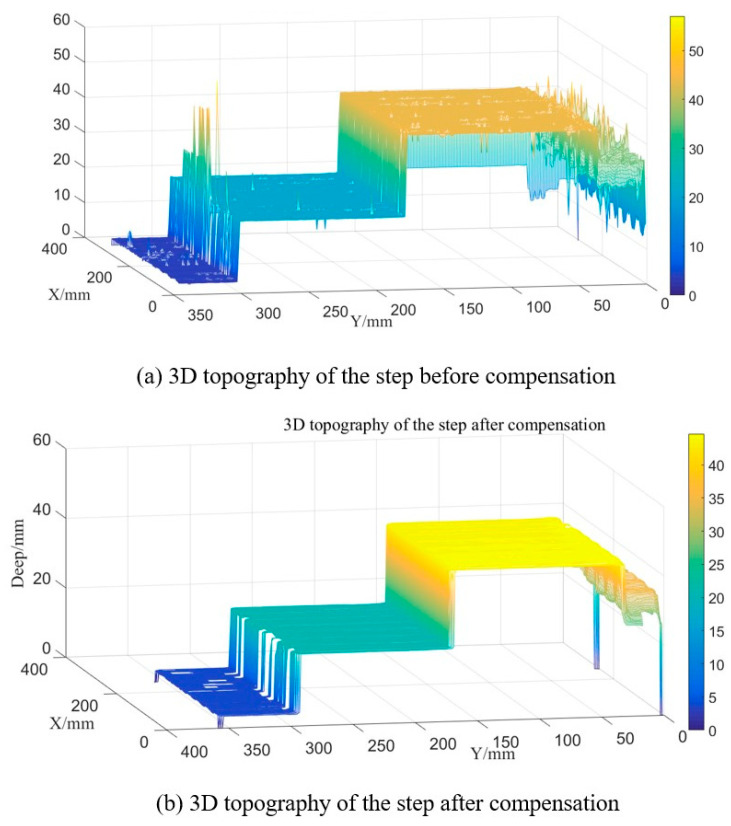
Three-dimensional step topography restoration map.

**Table 1 sensors-22-06845-t001:** Measurement precision of step distance before and after calibration (unit: mm).

	Standard Value	Before Compensation	After Compensation	Error before Compensation	Error after Compensation
Step1 and Step2	18.422	19.149	18.491	0.727	0.069
Step2 and Step3	13.258	12.537	13.163	−0.721	−0.095

## Data Availability

Data is contained within the article.

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
