# Peer review of "Phase Target-Based Calibration of Projector Radial Chromatic Aberration for Color Fringe 3D Measurement Systems"

_sensors, 2022, doi:10.3390/s22186845_

Round 1
Reviewer 1 Report
1) The literature review is too general. Moreover, the literature gap is not discussed.
2)The study should be compared with other reported work.
Author Response
Dear Reviewer,
We appreciate very much for the reviewers and editor’s comments on our manuscript, sensors-1856963. It has been revised according to these comments. In the revision notes, we respond to the comments point by point and show the accompanying manuscript revisions. The black content is the original text, the blue content is the response, and the red content is the modification. In addition, we have rewritten the article according the review comments and attached it at the end. We hope that the revised manuscript is now acceptable for publication!
Thanks very much for your attention to our manuscript.
Yours sincerely,
Yuzhuo Zhang

Reviewer 2 Report
A method for chromatic aberration correction is presented in this paper. A liquid crystal display with a holographic projection film as the phase target, the fringe phase displayed by the LCD regarded as the ideal one to compensate the phases projected by the projector. The technique route seems simple and feasible, but the content should be improved to meet the standards of Sensors publication.
Comments:
(1) In the introduction, the references about chromatic aberration correction should be supplied to support academic contribution of the proposed method.
(2) I am confused by the description of the section 3 and 4, and many improper sentences expression make the meaning unclear. So, the Sec. 3 and 4 should be rewrite and check carefully.
(3) The language should be polished by native English speakers.
(4) Images presented in the manuscript are a little blurred, so clarity of those images should be improved.
(5) The experiment presented in this paper is insufficient to support the conclusion, so measurement experiments should be added.
Author Response

(The authors gave the same response as above.)

Author Response

(The authors gave the same response as above.)

Round 2
Reviewer 1 Report
nothing.
Reviewer 2 Report
The revised paper is greatly improved, it can be accepted.